# Halogenated Dihydropyrrol-2-One Molecules Inhibit Pyocyanin Biosynthesis by Blocking the *Pseudomonas* Quinolone Signaling System

**DOI:** 10.3390/molecules27041169

**Published:** 2022-02-09

**Authors:** Theerthankar Das, Shekh Sabir, Ren Chen, Jessica Farrell, Frederik H. Kriel, Gregory S. Whiteley, Trevor O. Glasbey, Jim Manos, Mark D. P. Willcox, Naresh Kumar

**Affiliations:** 1Infection, Immunity and Inflammation, Charles Perkins Centre, Sydney Institute for Infectious Diseases, School of Medical Sciences, The University of Sydney, Sydney, NSW 2006, Australia; jessica.farrell@sydney.edu.au (J.F.); jim.manos@sydney.edu.au (J.M.); 2School of Chemistry, The University of New South Wales, Sydney, NSW 2052, Australia; s.sabir@student.unsw.edu.au (S.S.); r.chen@unsw.edu.au (R.C.); n.kumar@unsw.edu.au (N.K.); 3Whiteley Corporation, 19-23 Laverick Avenue, Tomago, NSW 2322, Australia; erik.kriel@whiteley.com.au (F.H.K.); greg.whiteley@whiteley.com.au (G.S.W.); trevor.glasbey@whiteley.com.au (T.O.G.); 4School of Optometry and Vision Science, The University of New South Wales, Sydney, NSW 2052, Australia; m.willcox@unsw.edu.au

**Keywords:** *P. aeruginosa*, pyocyanin, dihydropyrrol-2-one, LasR, PQS, quorum sensing

## Abstract

Quorum-sensing (QS) systems of *Pseudomonas aeruginosa* are involved in the control of biofilm formation and virulence factor production. The current study evaluated the ability of halogenated **dihydropyrrol-2-ones (DHP) (Br (4a), Cl (4b), and F (4c))** and a non-halogenated version (4d) to inhibit the QS receptor proteins LasR and PqsR. The DHP molecules exhibited concentration-dependent inhibition of LasR and PqsR receptor proteins. For LasR, all compounds showed similar inhibition levels. However, compound 4a (Br) showed the highest decrease (two-fold) for PqsR, even at the lowest concentration (12.5 µg/mL). Inhibition of QS decreased pyocyanin production amongst *P*. *aeruginosa* PAO1, MH602, ATCC 25619, and two clinical isolates (DFU-53 and 364707). In the presence of DHP, *P. aeruginosa* ATCC 25619 showed the highest decrease in pyocyanin production, whereas clinical isolate DFU-53 showed the lowest decrease. All three halogenated DHPs also reduced biofilm formation by between 31 and 34%. The non-halogenated compound 4d exhibited complete inhibition of LasR and had some inhibition of PqsR, pyocyanin, and biofilm formation, but comparatively less than halogenated DHPs.

## 1. Introduction

*Pseudomonas aeruginosa* is an opportunistic Gram-negative bacterium that is acknowledged as a critical pathogen by the World Health Organisation (WHO) [1]. *P. aeruginosa* is often multi-drug resistant, forms robust biofilms on biotic and abiotic surfaces, and is responsible for substantial morbidity and mortality in immunocompromised patients (especially those with Cystic Fibrosis or HIV infection) [2]. This bacterium is also the leading cause of nosocomial infections relating to ventilator-associated pneumonia, urinary tract, burns, wounds, and skin and soft tissue infections in hospitalized patients [2].

Biofilm formation and virulence factor production in *P. aeruginosa* is controlled by various signaling systems within *P. aeruginosa* including quorum sensing (QS) systems [3]. QS refers to population-dependent bacterial cell-to-cell signaling that can upregulate or downregulate certain genes that would favor bacterial survival, fitness, virulence factor production, pathogenicity in the host, biofilm formation, and resistance to antimicrobial agents. In *P. aeruginosa*, QS is driven by four signaling networks, LasI/LasR–AHL, RhlI/RhlR-BHL, PqsR–PQS, and IqsR-IQS [3]. LasR, RhlR, PqsR, and IqsR are transcriptional regulator receptors that bind to their respective autoinducer molecules (AHL, BHL, PQS, and IQS). These QS systems are arranged in hierarchies, with the LasI/R-AHL system often at the top of the hierarchy [3]. The LasI/R-AHL is involved in the regulation of production of protease and elastase; the RhlI/R-BHL system regulates elastase and rhamnolipid production; and the PqsR–PQS system regulates phenazine (predominantly pyocyanin) biosynthesis [4]. Pyocyanin is a virulence factor secreted by *P. aeruginosa* in the late exponential phase via PqsR–PQS mediated activation of *phzA1-G1* and phzA2-G2 operons [3,4].

The sputum of cystic fibrosis and bronchitis patients contains between 16.5 µg/mL and 27.3 µg/mL of pyocyanin, and exudates of infected wound dressing from patients with burn injuries contain up to 5.3 µg/g (mean 2.0 ± 2.3 µg/g) of pyocyanin [5,6]. Pyocyanin induces severe oxidative stress in host cells by producing reactive oxygen species such as hydrogen peroxide, depletes mammalian intracellular antioxidant (glutathione) levels, enhances the release of interleukin-8 (IL-8), and induces neutrophil-mediated tissue damage [7,8,9]. Pyocyanin can also induce premature cellular senescence (cell cycle arrest) in human diploid fibroblasts and, consequently, impairs wound repair [6]. Another crucial role of pyocyanin, discovered by Das et al., in 2015, is binding to DNA via intercalation with nitrogenous bases [10]. *P. aeruginosa* produces extracellular DNA (eDNA) as a major part of its biofilm formation [11]. Pyocyanin can bind to this eDNA to reduce its loss to the external environment [12]. Pyocyanin also aids in the establishment of the *P. aeruginosa* biofilm matrix and robust biofilms [12]. Pyocyanin can also facilitate electron transfer as the result of its binding to DNA, which might support the metabolic activity of bacterial subpopulations within biofilms [12].

Strategies that inhibit QS are being investigated as potential control mechanisms for various pathogens, considering the rise in bacterial antibiotic resistance and the development of superbugs. One strategy to combat QS is the synthesis of derivatives of lactone-based molecules, such as furanones and their analogues, that mimic the *N*-acyl homoserine lactones, which are the natural ligands in the *P. aeruginosa* LasR/LasI and RhlI/RhlR QS systems [13,14]. In this study, we synthesized halogenated dihydropyrrol-2-one (DHP) based homoserine lactone mimics. The structure of DHPs partly resembled AHL signaling molecules, and previous studies have demonstrated that DHPs possess both a six-membered pendant aromatic ring and a five-membered lactam ring and contain a conjugated exocyclic double bond, exhibit good LasR antagonist activity [15]. In our previous reports, we have synthesized several different analogues of DHP by modification at the nitrogen atom and substitution on the phenyl ring and on the exocyclic double bond, and these studies demonstrated that the exocyclic double bond and halogenated phenyl ring were essential for interfering with the LasR–AHL system [15,16]. In the current study, we tested these halogenated DHP compounds for their inhibition of the PQS QS by inhibiting PqsR receptor protein using a GFP-tagged *P. aeruginosa* PAO1 strain. We also tested these synthesized DHP compounds for their impact on pyocyanin production and biofilm formation with both laboratory and clinical isolates from infected wounds.

## 2. Results

### 2.1. Susceptibility of P. aeruginosa Isolates to Ciprofloxacin

Figure 1 shows the minimum inhibitory concentration (MIC) of *P*. *aeruginosa.* Strains PAO1–PQS, ATCC 25619 and the clinical isolate 364707 were sensitive to ≤1 µg/mL ciprofloxacin. The MIC for isolates MH602 and DFU-53 was 2 µg/mL. Data represent the average from three biological replicates.

### 2.2. DHP Compound Inhibition of LasR

Figure 2 details the expression of the LasR QS system of *P. aeruginosa* MH602, determined using the level of GFP fluorescence measured at Ex_485nm_ and Em_535nm_. In the controls, LasR-mediated GFP production peaked after 4 h and plateaued thereafter. This was also similar, although slightly delayed, when ethanol, the solvent for the DHP compounds, was present (Figure 2). Both halogenated and non-halogenated DHP compounds showed a concentration-dependent decrease in LasR receptor-mediated GFP production. In the presence of all four DHP compounds, the GFP produced as a function of the number of *P. aeruginosa* MH602 cells (OD_600nm_) was significantly decreased (*p* < 0.05). Increasing the concentration of any DHP compounds increased the inhibition of GFP production via the Las pathway. The greatest inhibition occurred with compounds **4a (Br)** and **4d**. At 50 µg/mL, all compounds showed complete inhibition of GFP fluorescence. Data represent the average from three biological replicates.

### 2.3. Inhibition of PqsR by DHP Compounds

Figure 3 shows the GFP fluorescence levels produced by activation of the PqsR QS system in *P. aeruginosa* PAO1–PQS. In the controls, PqsR-mediated production of GFP, began after one hour and peaked at 5 h. Both halogenated and non-halogenated DHP compounds showed a concentration-dependent decrease in PqsR receptor-mediated GFP over the 10 h of growth. Compared to the ethanol control, 12.5 µg/mL of compound **4a (Br)** reduced the production of PQS-mediated GFP after 5 and 6 h, and 25–50 µg/mL delayed the production of PQS-mediated GFP and reduced its production between 4 and 10 h. Compounds **4b (Cl)** and **4d** were less effective than **4a (Br)**, with 12.5 µg/mL delaying the production of PQS-mediated GFP but having no effect on GFP production after 4 h. Compound **4c (F)** delayed the production of PQS-mediated GFP at all concentrations tested, and 25 to 50 µg/mL reduced production between 2 and 10 h of growth. Data represent the average from three biological replicates.

### 2.4. Effects of DHPs Compounds on Planktonic Growth of P. aeruginosa

In general, DHP compounds showed minimal effect on the planktonic growth of *P. aeruginosa* strains, which is expected from quorum sensing inhibitors (QSIs). For all strains, there were slight reductions in growth (20%) at the highest concentration (50 µg/mL) of all DHP compounds. However, these differences in bacterial growth in the presence of DHP compounds were not statistically significant in comparison to controls. Growth data are representative of the average from three biological replicates (Figure 4A,B).

### 2.5. Effects of DHPs on Biofilm Formation by P. aeruginosa

*P. aeruginosa* strains PAO1–PQS, ATCC 25619, and DFU-53, grown in the presence of 50 µg/mL of any DHP compound, had a significant decrease in biofilm biomass (Figure 4C) of between 16 and 43% in comparison to control. However, the biofilm formation by strain DFU-53 was only affected by compound 4a. In the presence of solvent alone (1% ethanol), the biofilm formation of all *P. aeruginosa* strains was not significantly reduced (reduced by 8–17%). Biofilm biomass data are representative of the average from four biological replicates.

### 2.6. DHP Impedes Pyocyanin Production by P. aeruginosa Strains

Both halogenated and non-halogenated DHP compounds showed concentration-dependent decreases in pyocyanin production after 48 h of growth with all strains (Figure 5). For example, in the presence of DHPs, the production of pyocyanin from PAO1–PQS was reduced by 61–25% for 4a, 83–40% for 4b, 68–21% for 4c, and 90–59% for 4d in comparison to the controls, for isolates PAO1–PQS, MH602, ATCC 25619, and 364707. However, isolate DFU-53 from a foot ulcer showed reduced effects of the DHPs, but pyocyanin production was reduced by approximately 45% for compounds at 50 µg/mL. Refer Appendix A for pyocyanin production quantification. Data represent the average from *n* = 3 biological replicates.

### 2.7. Cytotoxicity Study of DHP Analogues **4a** and **4c** on Human Foreskin Fibroblasts (HFF-1)

Due to the overall performance of compounds **4a (Br)** and **4c (F)**, these were tested for their cytotoxicity. Resazurin cell metabolic assay and microscopy images confirmed that compounds **4a (Br)** at 12.5 and 25 µg/mL and **4c (F)** at 25 µg/mL produced no cytotoxicity on human foreskin fibroblast (HFF-1) cells after 24 h incubation. The cell morphology and confluence remained similar to the control, whereas the positive control of 100% DMSO produced severely disrupted cell morphology (Figure 6A). Cell viability was 100–94% with 12.5 and 25 µg/mL of the compounds, but reduced to 67–72% at 50 µg/mL (Figure 6B). When HFF-1 was exposed to the positive control DMSO, its viability decreased to <20%. Negative controls, where HFF-1 cells were treated with 0.9% NaCl, showed full confluence of cells and 100% viability. Cell viability is representative of the average from three biological replicates.

## 3. Discussion

Inhibition of QS with the aim of disrupting virulence factor production and biofilm formation is a key strategy to prevent bacterial pathogenicity. The current study evaluated QS inhibition by DHP analogues. DHP compounds had a concentration-dependent inhibitory effect on the AHL and PQS QS systems of *P. aeruginosa*. When treated with the DHP analogues, there was a reduction in the production of GFP that was linked to LasR and PqsR receptor proteins, as well as the production of phenazine and biofilm. LasR binds its cognate ligand AHL/HSL to trigger its activation and the transcription of various genes, including those associated with the production of virulence factors [3,4]. PqsR binds to the PQS signaling molecule 2-heptyl-3-hydroxy-4-quinolone and then transcribes phenazine synthesis genes (*PhzA1-G1* and *A2-G2*) [3]. The QS systems in *P. aeruginosa* are interconnected with the PQS system being controlled by the LasR–AHL/HSL system [17]. Therefore, inhibition in LasR receptor proteins leads to a decline in PqsR receptors and, ultimately, a reduction in pyocyanin production (Figure 7).

In general, the differences in electronegativity and hydrophobicity of DHP compounds driven by the presence of halogens (Br, Cl, and F) did not significantly affect pyocyanin production or biofilm formation, but did affect activation of the PqsR receptor. Compound **4a**, containing bromine, inhibited the production of GFP by PqsR between two-(at 12.5 µg/mL) and eight-(at 50 µg/mL) fold. Compound **4b**, containing chlorine, reduced the production of GFP by PqsR by one- to two-fold, and compound **4c**, containing fluorine, reduced production by one- to nine-fold. Studies have shown that pyocyanin production is directly related to virulence and severity of infection [18,19], and so inhibition of pyocyanin would likely reduce pathogenicity in a host.

Another role of pyocyanin is its ability to increase the stability of the *P. aeruginosa* biofilm matrix by enhancing extracellular DNA production [20,21] and binding with extracellular DNA to form a stable biofilm [10]. In the current study, DHP compounds produced an approximately 30% decrease in biofilm biomass. This decrease may have been due to the reduction in pyocyanin affecting the viscosity of the biofilm matrix. The biofilm matrix viscosity is an essential factor that helps biofilms to be resistant to shear stress and antibacterial agents [22,23]. In the case of *P. aeruginosa,* pyocyanin binding to DNA enhances the viscosity of the eDNA biofilm matrix [10,22]. A reduction in pyocyanin may weaken the architecture of the biofilm and thus allow the biofilm to slough off from surfaces more easily [10,24].

All DHP compounds produced a drastic decrease in LasR- and PqsR-mediated production of GFP and a significant reduction in pyocyanin, and biofilm formation. Also, compounds 4a and 4c were not cytotoxic as per the ISO 10993-5:2009—direct contact method, on the HFF-1 cell line [25]. Fibroblasts are crucial cell lines for wound healing. Data from the current study indicate that when administrated responsibly or at a moderate concentration of 25 µg/mL, DHPs have the potential to reduce *P. aeruginosa* pathogenicity by minimizing virulence factor production and impairing biofilm formation, with limited cytotoxicity to human cells. Previous studies have shown that DHP compounds combined with antibiotics increase the susceptibility of *P. aeruginosa* to those antibiotics [26,27], and this may further increase their utility. However, a study has demonstrated that recurrent exposure of *P. aeruginosa* PAO1 biofilms to a combination of the furanone (C-30), an inhibitor of the LasR/I system, and tobramycin resulted in an increased resistance triggered by mutation or deletion in antibiotic resistance genes (*mexT*, *fusA1*, and *parS*) [27]. The current compounds should be analyzed similarly to determine whether they are also associated with increased resistance to conventional antibiotics.

## 4. Materials and Methods

### 4.1. Synthesis

All the DHP analogs have been synthesized in sequential steps by previously reported protocols. Synthesis commences through a condensation reaction between commercially available halogenated derivatives of phenylacetone (1) and glyoxylic acid in the presence of phosphoric acid, which gives a lactone intermediate (2). This lactone is converted into the corresponding lactam analog (3) in two steps, including heating with thionyl chloride followed by reaction with aqueous ammonia. The final step involves dehydration of the 5-hydroxy-lactam compound by phosphorous pentoxide (P_2_O_5_), which yields the target 5-methylene-4-phenyl- 1,5-dihydro-2H-pyrrol-2-ones (4) (Figure 8) [28,29].

### 4.2. P. aeruginosa Isolates, Culture and Growth Conditions

*P. aeruginosa* strains used in this study included *P. aeruginosa* MH602 *las*B::*gfp*(ASV) for LasR analysis and PAO1 *pqs*A::*gfp* for PqsR analysis, ATCC 25619, and clinical isolates DFU-53 (high-risk foot clinic, Liverpool Hospital, Sydney, NSW, Australia) and 364707 (a head wound isolate from the Microbiology Department, Royal Prince Alfred Hospital, Sydney, NSW Australia). All clinical isolates were patient de-identified before being gifted to us from the hospitals. Clinical bacteria from patients were previously isolated and characterized by qualified pathologists from the respective hospitals based on procedures detailed in *ASM Clinical Microbiology Procedures Handbook—4th Edition, ASM Press*
*(edited by Amy L. Leber)*. None of the authors were involved in the isolation or characterization of clinical isolates from patients. Bacterial isolates, once received in our research laboratory, were immediately grown in a full-strength (100%) Tryptone Soy Broth (Oxoid, ThermoFisher, Sydney, NSW, Australia) in a test tube at 37 °C, 150 rpm for 24 h. Following 24 h of growth, the bacterial stock was prepared by mixing 1.2 mL of bacterial culture with 400 µL of DMSO (100%), and stored in 2 mL of the cryogenic vials (Corning, Sigma-Aldrich, Sydney, NSW, Australia) at −80 °C. To note: the final concentration of DMSO in the bacterial stock is 25%. *P. aeruginosa* isolates from the frozen stock were plated as required onto Tryptone Soy agar (Oxoid) and grown for 24 h by incubating at 37 °C. Further growth was initiated by striking a colony from the Tryptone Soy agar plate into the 5 mL of Muller–Hinton Broth (MHB; Oxoid) for 24 h at 37 °C, with orbital shaking at 150 rpm for all experiments.

### 4.3. P. aeruginosa Isolates’ Susceptibility to Ciprofloxacin

The antibiotic susceptibility of all *P. aeruginosa* strains used in this study was tested aganist ciprofloxacin. A stock of ciprofloxacin (100 µg/mL) was prepared by dissolving it in MHB. *P. aeruginosa* strains were grown for 24 h at 37 °C, 150 rpm in 5 mL MHB, as mentioned above. After 24 h of growth, each *P. aeruginosa* culture was diluted (by resuspending in MHB to a density of 0.15 ± 0.04 (OD_600nm_)), either in the presence or absence of ciprofloxacin (1, 2, and 5 µg/mL). Then, 200 µL of the dilute *P. aeruginosa* culture (in the presence or absence of ciprofloxacin) was placed in the 96-well plates and the growth was recorded by measuring the absorbance at OD_600nm_ at 0 after 24 h incubation at 37 °C. Any increase in growth at 24 h was determined by subtracting the absorbance readings obtained at 0 h.

### 4.4. LasR and PqsR Detection and Growth Quantification

*P. aeruginosa* MH602 *las*B::*gfp*(ASV) and PAO1 *pqs*A::*gfp* isolates were grown overnight in MHB, supplemented with 30 µg/mL gentamicin at 37 °C, 150 rpm [30,31]. Additionally, 500 µL of bacterial culture (grown overnight in MHB) was further diluted in M9 minimal salt media (Sigma-Aldrich, Sydney, NSW, Australia) to a total volume of 5 mL, either in the presence or absence of DHP compounds (12.5, 25, and 50 µg/mL). For untreated control and to test the impact of solvent, PBS and ethanol (equivalent to respective DHP concentrations), respectively, were used in place of DHP compounds. The final bacterial cell density in 5 mL of diluted culture was approximately 0.1 ± 0.025 at OD_600nm_, measured using a plate reader (Tecan infinite M1000 Pro, ThermoFisher, Australia). M9 minimal media was chosen due to its low autofluorescence in comparison to MHB and Tryptone soy broth. Finally, 200 µL of diluted M9 minimal salt medium was dispensed into 96-well plates (Corning Corp., Corning, NY, USA) and incubated at 37 °C with orbital shaking at 150 rpm. Production of GFP via the receptor proteins LasR and PqsR and bacterial growth was monitored every hour for up to 10 h by measuring fluorescence at Ex_485nm_ and Em_535nm_ and absorbance at OD_600nm_ for LasR and PqsR and bacterial growth, respectively. Bacterial growth (OD_600nm_) was further measured after 24 h and 48 h of treatment. To quantify LasR and PqsR expression, the GFP fluorescence values measured at every time point were divided by their respective OD_600nm_ absorbance. A graph was plotted as GFP/OD vs. time in hours.

### 4.5. P. aeruginosa Biofilm Biomass Quantification Using Crystal Violet

For assessing biofilm biomass, *P. aeruginosa* was grown as described above. The bacterial culture (200 µL) was aliquoted in 96-well plates at a density of 0.1 ± 0.025 (OD_600nm_). The bacteria were incubated for 48 h at 37 °C with 150 rpm orbital shaking in M9 minimal media containing 10% MHB, in the presence or absence of DHP. After 48 h, the supernatant was removed and the biofilm was washed once with phosphate-buffered saline (PBS; 137 mM NaCl, 2.7 mM KCl and 10 mM phosphate, pH 7.41, POCD Healthcare, North Rocks, NSW, Australia) to remove any loosely bound bacterial cells. The attached biofilms were then treated with 0.01% crystal violet dye and incubated for 60 min at 37 °C and 150 rpm. After 60 min, excess dye was removed by washing with PBS three times. The plates were then dried for 20 min at 37 °C, followed by the addition of 200 µL of 80% ethanol and incubation for 30 min at 37 °C, with orbital shaking at 150 rpm to dissolve the dye from the biofilm biomass. Aliquots (100 µL) of dye were transferred to a new 96-well plate, to which 100 µL of PBS was added to make up a total volume of 200 µL in each well. The plates were then analyzed for biofilm biomass by measuring absorbance at OD_550nm_ using a plate reader. As mentioned in Section 4.4, the controls were growth in the absence of DHP and in the presence of the equivalent amount of DHP solvent (ethanol).

### 4.6. Quantification of Pyocyanin Production

All *P. aeruginosa* strains were grown overnight in MHB, except for PAO1 and MH602, which were grown in MHB supplemented with 30 µg/mL gentamicin, at 37 °C with orbital shaking at 150 rpm. Bacterial cultures grown in MHB were diluted to a total of 5 mL of M9 minimal salt media in presence or absence of DHP 12.5, 25, and 50 µg/mL and PBS or DHP ethanol solvent to a density of 0.1 ± 0.025 (OD_600nm_) and incubated for 48 h at 37 °C with 150 rpm orbital shaking. After 48 h, cultures were transported in a sterile 15 mL Falcon tube and centrifuged at 4500 g for 10 min at 10 °C. After centrifugation, the supernatant was removed and pyocyanin was extracted as described previously [32]. In brief, the supernatant was treated with chloroform to a ratio of 1:0.6 (i.e., to every 1 mL of supernatant, 600 µL of chloroform added). The mixture was immediately vortexed for 5 s and centrifuged for a further 10 min at 4500× *g* at 10 °C. The bottom blue layer was pipetted into sterile 15 mL Falcon tubes and treated with 0.1 M HCl at a ratio of 1:0.5 (i.e., 500 µL of 0.1 M HCl was added to every 1 mL of blue solution). The mixture was vortexed for 5 s, followed by centrifugation for 10 min at 10 °C and 4500× *g*. An aliquot (200 µL) of the final pink top layer composed of acidified pyocyanin was pipetted into wells of a 96-well plate and quantified for pyocyanin by measuring absorbance at OD_520nm_ using a Tecan Infinite M1000 Pro plate reader.

### 4.7. Analysis of DHP Cytotoxicity to Human Fibroblast Cells

After analysis of the results, compounds **4a** and **4c** showed good activities in all assays and thus were chosen to assess their impact on human fibroblast cell lines HFF-1. Cytotoxicity was assessed based on the British Standard ISO 10993-5:2009—direct contact assay [25]. HFF-1 (passage #14–16) were cultured in DMEM medium supplemented with fetal bovine serum (12% *v/v*), penicillin (100 IU/mL), and streptomycin (100 μg/mL). HFF-1 cells were grown at 37 °C in a 5% (*v/v*) CO_2_ and harvested at 90% confluence using 0.12% *v/v* trypsin–EDTA. Cells were collected by first quenching Trypsin 1:1 *v/v* with supplemented media and transferred to conical centrifuge tubes, followed by centrifugation (5 min, 4500× *g*, 20 °C). The supernatant was aspirated, and the cell pellet was suspended in supplemented DMEM media for further experiments.

After harvesting, HFF-1 cells were plated to a density of 105 cells/mL into six-well plates (Corning Corp., Corning, NY, USA) and allowed to incubate at 37 °C in a 5% (v/v) CO2 atmosphere, with the medium being exchanged every alternative day until a confluence of 95–100% of cells (verified using a light microscope) was achieved. At 95–100% confluence, the medium was discarded and cells were supplemented with 1 mL of the growth medium, followed by placement of a freshly made filter disc containing 50 µL of 4a (12.5, 25, and 50 µg/mL) or 4c (25 and 50 µg/mL) in the center of the wells. The filter disc was prepared by adding 50 µL of test compounds directly onto 6 mm antibiotic filter paper discs and incubating them at room temperature for 2 h to allow the discs to absorb test compounds. The plates containing HFF-1 and the filter disc were then incubated for 24 h at 37 °C in a 5% (v/v) CO2 atmosphere. After 24 h, the filter disc and the cell growth media were removed and replaced with 1 mL sterile PBS. The cell morphology of the HFF-1 cells and their confluence were assessed using a light microscope.

Quantification of HFF-1 cell metabolic activity (a cell viability assay) was analyzed by adding to the cells 50 µL of 0.05% resazurin solution (Sigma-Aldrich, Sydney, NSW, Australia; prepared in sterile MilliQ water) to 1 mL PBS and incubating them for 24 h at 37 °C in a 5% (*v/v*) CO_2_. After 24 h, the fluorescence of the HFF-1 cells was measured at Ex_544nm_ and Em_590nm_ using a plate reader. For the controls, a filter disc containing 50 µL 0.9% *w/v* NaCl and another containing 50 µL 100% DMSO (a positive control that causes cells death) were used. Based on ISO 10993-5:2009 [25], DHP compounds were considered non-cytotoxic if the viability of cells remained above 80% (grade 0) and mild if cell lysis is not more than 50% (grade 2). Experiments were conducted in biological triplicates.

### 4.8. Statistical Analysis

For all statistical analysis, GraphPad Prism (https://www.graphpad.com/quickcalcs/ttest1.cfm (accessed on 23 December 2021)) was used for conducting two-tailed unpaired *t-*tests in order to determine the *p*-values. The data presented are of the mean ± standard deviation (SD), with *p*-values < 0.05 deemed as significant.

## 5. Conclusions

Developing compounds that inhibit QS is a strategy to reduce the production of virulence factors as well as biofilm formation (and hence limit antibiotic use) to combat bacterial infections. DHPs have a significant impact on isolates of *P. aeruginosa* by inhibiting virulence factor (pyocyanin) production and biofilm formation whilst having low cytotoxicity to human fibroblasts. This current study needs to be extended to elucidate the resistance potential of *P. aeruginosa* after repeat exposure to DHP and whether exposure to DHPs can change their antibiotic susceptibility profile. Further, an exploration of how DHPs affect transcription of the genome of *P. aeruginosa* would provide valuable additional information.

## Figures and Tables

**Figure 1 molecules-27-01169-f001:**
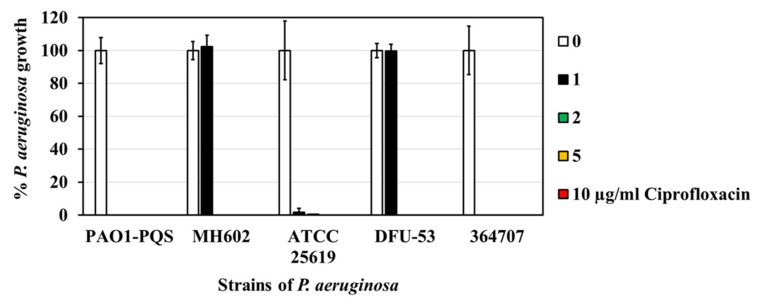
The susceptibility of *P. aeruginosa* isolates to ciprofloxacin.

**Figure 2 molecules-27-01169-f002:**
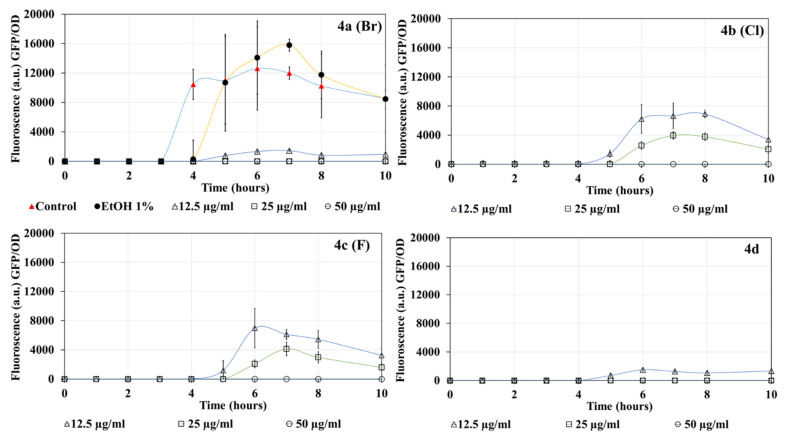
Effects of DHP compounds (12.5–50 µg/mL) on the production of GFP mediated through LasR in *P. aeruginosa* MH602.

**Figure 3 molecules-27-01169-f003:**
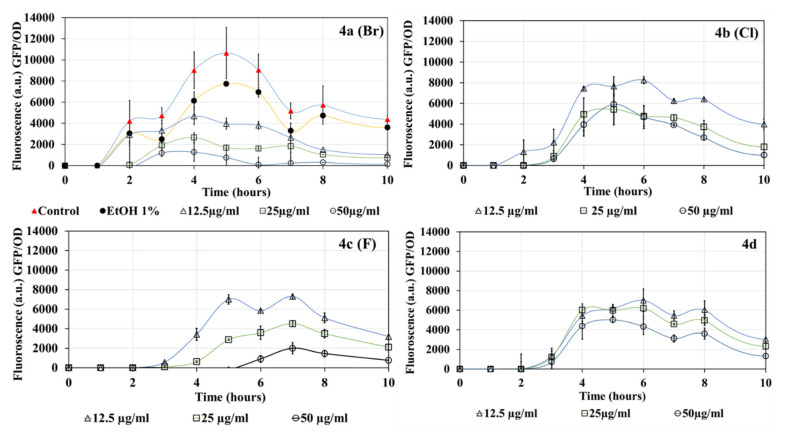
Effects of DHP compounds on PQS-linked GFP production by *P. aeruginosa* PAO1.

**Figure 4 molecules-27-01169-f004:**
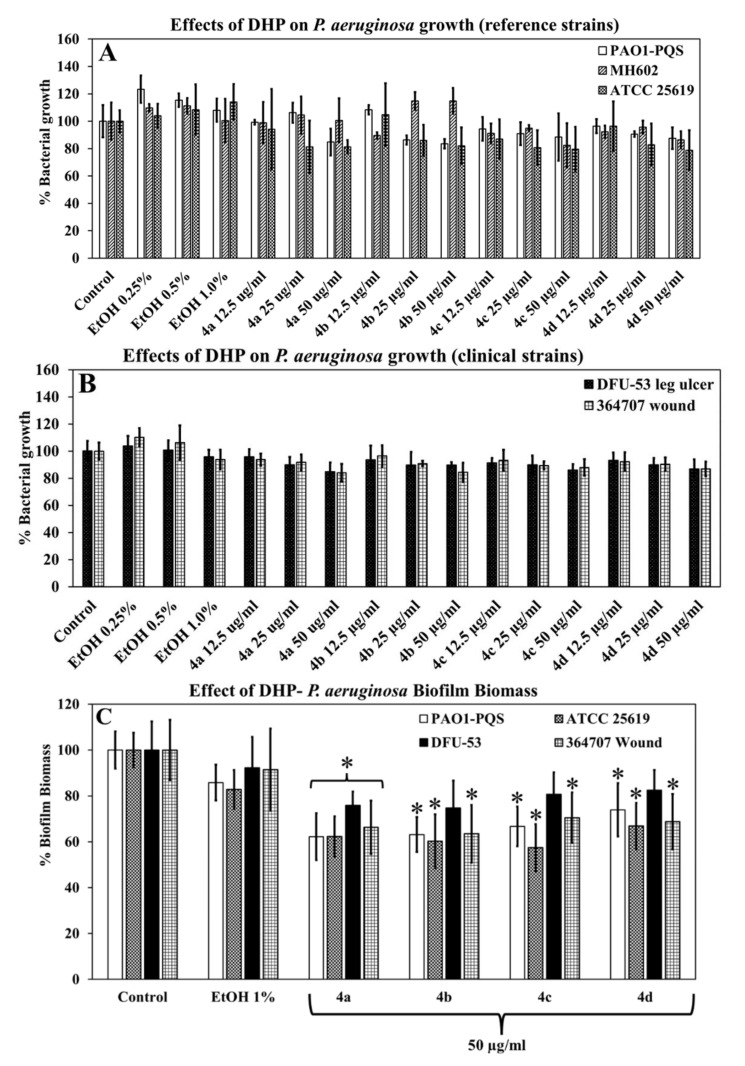
(**A**,**B**) Effect of DHP compounds on *P. aeruginosa* planktonic growth. (**C**) Effect of DHP compounds on *P. aeruginosa* biofilm biomass formation. * Indicates differences are statistically significant *p* < 0.05.

**Figure 5 molecules-27-01169-f005:**
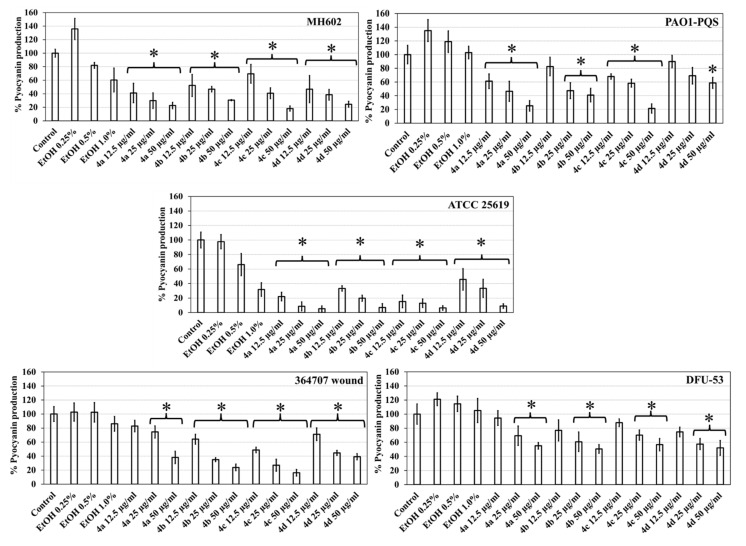
Inhibition of pyocyanin production by *P. aeruginosa* by DHP compounds. * Indicates differences are statistically significant. *p* < 0.05.

**Figure 6 molecules-27-01169-f006:**
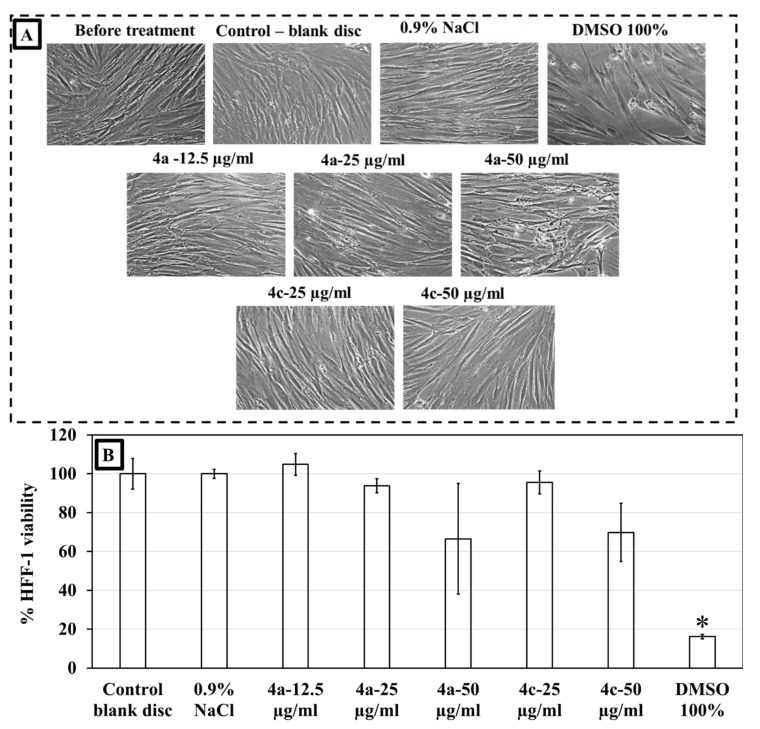
Cytotoxic effects of DHP **4a (Br)** and **4c (F)** and DMSO on HFF-1 cells. (**A**) Microscopic images showing morphology of HFF-1 cells and (**B**) Metabolic viability of HFF-1 cells. * Indicates differences were statistically significant *p* < 0.05.

**Figure 7 molecules-27-01169-f007:**
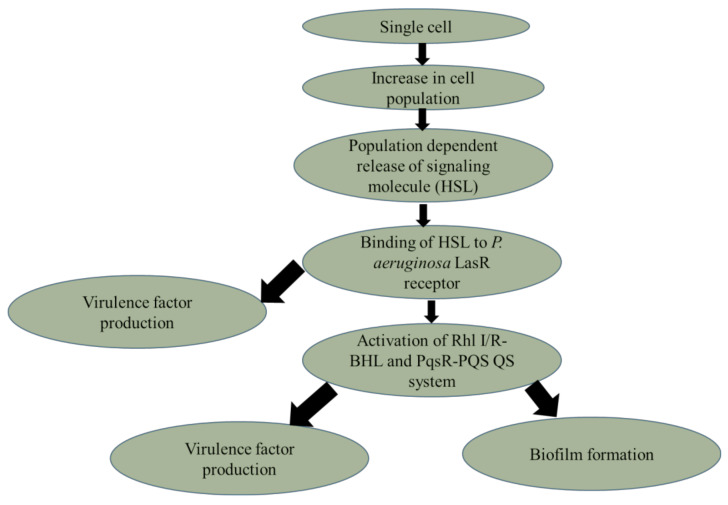
Schematic showing how *P. aeruginosa* QS systems result in virulence factor production and biofilm formation. Binding of AHL/HSL to LasR triggers transcription of Rhl I/R and PqsR and the production of their ligands BHL and PQS. These QS systems then coordinated the biosynthesis of virulence factors such as pyocyanin and the formation of biofilm.

**Figure 8 molecules-27-01169-f008:**
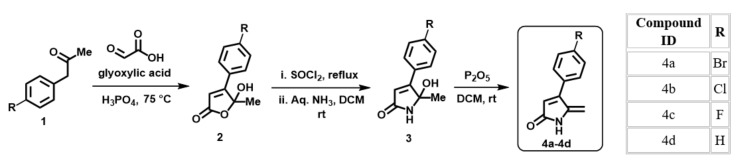
Step-by-step synthesis of halogenated dihydropyrrol-2-one molecules. Compound ID and respective halogen atom labelled as “R”.

## Data Availability

Not applicable.

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
