# Peer review of "Halogenated Dihydropyrrol-2-One Molecules Inhibit Pyocyanin Biosynthesis by Blocking the Pseudomonas Quinolone Signaling System"

_molecules, 2022, doi:10.3390/molecules27041169_

Round 1

Reviewer 1 Report

The paper explored the quorom sensing inhibition of P aeruginosa biofilm by arylated pyrrolidinone derivatives via investigation of their inhibitory properties on two quorom-sensing receptor proteins namely LasR and PqsR. I find the paper remarkably interesting because it presented an examination of the mechanism of derivatives inspired from QS inhibitory natural products. While the biological assessment for QS inhibition was thoroughly studied, I find the chemical side of the study lacking justification.

  1. The authors failed to explain/justify in their text (both intro and discussion) the basis for choosing only this set of derivatives. I wonder why they are only reduced to producing only this limited number of congeners when the synthesis is easy/straightforward.
  2. In all assays, data for positive drug control are missing for a meaningful assessment of QS inhibitory potency. Or is it ethanol that is the positive control used in their experiments?
  3. The paper was not prepared carefully prior to submission. There are numerous errors in formats like presentation of compound citations, organism nomenclature, etc. The Figures were not carefully drawn and cited.

Author Response

Reviewer 1

 Comments and Suggestions for Authors

The paper explored the quorom sensing inhibition of P aeruginosa biofilm by arylated pyrrolidinone derivatives via investigation of their inhibitory properties on two quorom-sensing receptor proteins namely LasR and PqsR. I find the paper remarkably interesting because it presented an examination of the mechanism of derivatives inspired from QS inhibitory natural products. While the biological assessment for QS inhibition was thoroughly studied, I find the chemical side of the study lacking justification.

Reply: We thank the reviewer for their comments and suggestion. We have considered the reviewer’s suggestions and made all changes. All changes in the manuscript are highlighted in yellow and all line numbers below are as per the revised manuscript.

  1. The authors failed to explain/justify in their text (both intro and discussion) the basis for choosing only this set of derivatives. I wonder why they are only reduced to producing only this limited number of congeners when the synthesis is easy/straightforward.

Reply: Thanks for pointing this out. As we are continuously working on this scaffold, in our previous study we found that the halogenated DHP compounds are most active in the LasR quorum-sensing system. Therefore, in this study, we have evaluated the effect of these halogen-containing DHP compounds on the PqsR quorum-sensing system. As suggested, we added relevant text in the introduction section. Lines 74-78.

  1. In all assays, data for positive drug control are missing for a meaningful assessment of QS inhibitory potency. Or is it ethanol that is the positive control used in their experiments?

Reply: As mentioned in the results, figures, and methods, ethanol was used as a solvent to dissolve all DHP compounds. Hence, ethanol 0.25-1% (maximum) was used to study the impact of solvent (positive control) and in addition, untreated control (that do not have DHP compounds or solvent) in experiments throughout the study. For more clarity, we have now clearly mentioned it in the method sub-section 4.3, 4.4, and 4.5.

  1. The paper was not prepared carefully prior to submission. There are numerous errors in formats like presentation of compound citations, organism nomenclature, etc. The Figures were not carefully drawn and cited.

Reply: As suggested, we have made significant changes throughout the manuscript including English grammar, changes in figures, text style, font size, the nomenclature of the organism compound citations, etc.

Reviewer 2 Report

  1. The authors should take the manuscript for English editing as many grammatical errors make reading difficult.
  2. From the manuscript title, Molecules Inhibit Pyocyanin Biosynthesis by Blocking the Pseudomonas Quinolone Signaling System. Therefore, authors must add susceptibility of Pseudomonas strain to Quinolone antibiotics group in results.
  3. AbbreviationWhen used for the first time you write complete and abbreviationbetween brackets.     
  4. Line 22, 23 Inhibition of QS decreased pyocyanin production amongst both laboratory 22 (PAO1, MH602, and ATCC) and clinical wound isolates (DFU-53 and 364707). Correct to Inhibition of QS decreased pyocyanin production amongst aeruginosa PAO1, MH602, ATCC 25619, and two clinical isolates (DFU-53 and 364707)
  5. Line 23, 24 In the presence of DHP, the ATCC isolate showed the highest decrease in pyocyanin production 24 whereas DFU-53 showed the least impact. Correct to In the presence of DHP, the aeruginosa ATCC 25619 showed the highest decrease in pyocyanin production whereas clinical isolate DFU-53 showed the least impact.
  6. Line 25, 26 DHPs also reduced biofilm formation by between 31 to 34% for all three compounds. Correct to DHPs also reduced biofilm formation by between 31 to 34% for all three halogenated compounds
  7. Line 26, 27, 28 The non-halogenated version 4d exhibited complete inhibition of LasR and had some inhibition of PqsR, pyocyanin, and biofilm formation but comparatively less than halogenated DHPs. Correct to The non-halogenated compound 4d exhibited complete inhibition of LasR and had some inhibition of PqsR, pyocyanin, and biofilm formation but comparatively less than halogenated DHPs
  8. Pseudomonas aeruginosa write italic Pseudomonas aeruginosa in all manuscript text.
  9. Line 43 and resistance against antimicrobial agents correct to and resistance to antimicrobial agents
  10. Line 93 Increasing the concentration of any DHP increased the inhibition of GFP production via the Las pathway. Correct to, Increasing the concentration of any DHP compounds increased the inhibition of GFP production via the Las pathway.
  11. Line 96 Figure 1. DHP compounds inhibit LasR-mediated production of GFP in aeruginosa MH602. Both halogenated and non-halogenated DHP compounds showed a concentration-dependent decrease in LasR receptor-mediated GFP production over 10 hours of growth. Peak 98 GFP production for all growth conditions occurred between 4-7 hours. At the 4-7 hours point even 99 in lowest concentration (12.5 μg/ml) of all DHPs produced a significant decrease (P < 0.05) in fluorescence compared to the control was observed. At 50 μg/ml, all compounds showed a complete 101 decrease in GFP fluorescence at 4-7 hours point. Data represent the average from three biological 102 replicates. Correct to  Figure 1. Showed effects of DHP compounds on GFP production in P. aeruginosa MH602, and write  P. aeruginosa with the same font size until matched with line. Add paragraph  Both halogenated and non-halogenated DHP compounds showed a concentration-dependent decrease in LasR receptor-mediated GFP production over 10 hours of growth. Peak 98 GFP production for all growth conditions occurred between 4-7 hours. At the 4-7 hours point even 99 in lowest concentration (12.5 μg/ml) of all DHPs produced a significant decrease (P < 0.05) in fluorescence compared to the control was observed. At 50 μg/ml, all compounds showed a complete 101 decrease in GFP fluorescence at 4-7 hours point. Data represent the average from three biological 102 replicates.  To results
  12. Line 118 Figure 2. DHP compounds inhibit GFP produced via the PqsR receptor protein in aeruginosa PAO1. Both halogenated and non-halogenated DHP compounds showed a concentration 119 dependent decrease in PqsR receptor mediated GFP over the 10 hours of growth. The GFP fluorescence peak for all growth conditions was between 4-7 hours point. At 4-7 hours point, even in the 121 lowest concentration of DHP (12.5 μg/ml), a decrease in fluorescence was recorded in comparison 122 to the control. Whereas, at 25 and 50 μg/ml all compounds showed a significant decrease (P < 0.05) 123 in GFP fluorescence at 4-7 hours point. Data represent the average from three biological replicates. . Correct to  Figure 2. Showed effects of DHP compounds on  GFP production in P. aeruginosa PAO1. Added paragraph Both halogenated and non-halogenated DHP compounds showed a concentration 119 dependent decrease in PqsR receptor mediated GFP over the 10 hours of growth. The GFP fluorescence peak for all growth conditions was between 4-7 hours point. At 4-7 hours point, even in the 121 lowest concentration of DHP (12.5 μg/ml), a decrease in fluorescence was recorded in comparison 122 to the control. Whereas, at 25 and 50 μg/ml all compounds showed a significant decrease (P < 0.05) 123 in GFP fluorescence at 4-7 hours point. Data represent the average from three biological replicates to results
  13. The authors used Green Fluorescent Protein (GFP) to determine the inhibition level of LasR and PqsR with the same methodology. How do authors discriminate between LasR and PqsR?
  14. In figure3A growth of effect of W1- aeruginosa growth reference strain growth correct to  Figure3A effects of DHP on P. aeruginosa growth (reference strain)
  15. In figure 3B growth of effect of W1- aeruginosa growth reference strain growth correct to  Figure3B effects of DHP on P. aeruginosa growth (clinical strain)
  16. Line 125 DHP compounds have minimal impact on P. aeruginosa growth correct to Assess effects DHP compounds on aeruginosa.
  17. Title figure 3 correct to Figure 3 Effects of DHP compounds on aeruginosa growth and biofilm biomass.  Paragraph  DHP compounds at any concentration 140 did not significantly affect the growth of any strain (A and B). In presence of DHP compounds, biofilm formation was 141 significantly reduced for all isolates except for DFU-53 which was only significantly affected by compound 4a after 48 142 hours of biofilm formation (C). Growth data is representative of the average from three biological replicates. Biofilm bio-143 mass data is representative of the average from four biological replicates * indicates differences are statistically significant 144 P < 0.05. added to results     
  18. Line 146 Biofilm formation is hindered in the presence of DHP correct to Effects of DHP on biofilm formation by aeruginosa
  19. Line 154 DHP impedes pyocyanin production of P. aeruginosa isolates correctly to DHP impedes pyocyanin production by P. aeruginosa strains
  20. Line 286 clinical strains correct to clinical isolates
  21. Title figure 5 Figure 4. DHP compounds inhibit pyocyanin production by aeruginosa isolates. Both 173 halogenated and non-halogenated DHP compounds showed concentration-dependent decreases 174 in pyocyanin production after 48 hours of growth. At 50 μg/ml all compounds showed a signifi-175 can’t decrease. Data represent the average from n=3 biological replicates. * Indicates differences are 176 statistically significant P < 0.05. correct to Figure 4. Showed effects of DHP on pyocyanin production by P. aeruginosa strains and write with one font size. Paragraph  Both halogenated and non-halogenated DHP compounds showed concentration-dependent decreases in pyocyanin production after 48 hours of growth. At 50 μg/ml all compounds showed a significant decrease. Data represent the average from n=3 biological replicates. * Indicates differences are 176 statistically significant P < 0.05 added to results
  22. Line 178 Bromophenyl-DHP analogue (4a) shows concentration-dependent cytotoxicity to HFF-1 cell correct to Cytotoxicity Study of -DHP analogue (4a) on human foreskin fibroblast (HFF-1) cell.
  23. Paragraph Microscopy images showing HFF-1 cell 190 morphology and confluence at 24 hours (A). At the low and moderate concentrations (12. 5 and 191 μg/ml), 4a was not cytotoxic but at 50 μg/ml the viability of HFF-1 reduced to approximately 67% 192 (B). Cell viability is representative of the average from three biological replicates. * Indicates differences were statistically significant P < 0.05. deleted from the title of figure 5 and added to results.
  24. Line 289 All clinical isolates were de-identified before being gifted to us from the hospitals. None of the authors were involved in the isolation or characterization of clinical isolates from 290 patients. The must be mentioned who is responsible for the identification of bacterial isolates and the method used.
  25. Line 291 authors mention, All isolates were stored at -80ºC in 25% DMSO and used as necessary by plating, How bacteria are stored in 25% DMSO? Mention source this is information?
  26. In the statical analysis part, the authors said, For all statistical analysis, Graphpad Prism 384 was used for two-tailed unpaired T-test to determine the P-values. Why did not the authors perform an ANOVA analysis to compare the various compound at the level of each effect (growth quantification assay, biofilm biomass quantification, pyocyanin production, etc.) and therefore choose the appropriate molecule to employ the cytotoxicity experiment?

Author Response

Reviewer 2

 Comments and Suggestions for Authors

Reply: We thank the reviewer for their comments and suggestion. We have considered the reviewer’s suggestions and made all changes. All changes in the manuscript are highlighted in yellow.

  1. The authors should take the manuscript for English editing as many grammatical errors make reading difficult.

Reply: Thanks for the advice, we have now checked and edited the text for grammatical and other errors throughout the manuscript.

  1. From the manuscript title, Molecules Inhibit Pyocyanin Biosynthesis by Blocking the Pseudomonas Quinolone Signaling System. Therefore, authors must add susceptibility of Pseudomonas strain to Quinolone antibiotics group in results.

Reply:  We agree with reviewer comments, hence we have now included the Ciprofloxacin (fluoroquinolone) susceptibility study for all P. aeruginosa isolates used in this study. Refer to Figure 1 and corresponding text in methods sub-section 4.3 and results in sub-section 2.1.

  1. Abbreviation When used for the first time you write complete and abbreviation between brackets.   

Reply:   We have now checked and edited for abbreviations throughout the manuscript.

  1. Line 22, 23 Inhibition of QS decreased pyocyanin production amongst both laboratory 22 (PAO1, MH602, and ATCC) and clinical wound isolates (DFU-53 and 364707). Correct to Inhibition of QS decreased pyocyanin production amongst aeruginosa PAO1, MH602, ATCC 25619, and two clinical isolates (DFU-53 and 364707)

Reply: Changes made as suggested by the reviewer. Lines 21-23.

  1. Line 23, 24 In the presence of DHP, the ATCC isolate showed the highest decrease in pyocyanin production 24 whereas DFU-53 showed the least impact. Correct to In the presence of DHP, the aeruginosa ATCC 25619 showed the highest decrease in pyocyanin production whereas clinical isolate DFU-53 showed the least impact.

Reply: Changes made as suggested by the reviewer. Lines 23-25.

  1. Line 25, 26 DHPs also reduced biofilm formation by between 31 to 34% for all three compounds. Correct to DHPs also reduced biofilm formation by between 31 to 34% for all three halogenated compounds

Reply: Changes made as suggested by the reviewer. Line 25-26.

  1. Line 26, 27, 28 The non-halogenated version 4d exhibited complete inhibition of LasR and had some inhibition of PqsR, pyocyanin, and biofilm formation but comparatively less than halogenated DHPs. Correct to The non-halogenated compound 4d exhibited complete inhibition of LasR and had some inhibition of PqsR, pyocyanin, and biofilm formation but comparatively less than halogenated DHPs

Reply: Changes made as suggested by the reviewer. Line 26-28.

  1. Pseudomonas aeruginosa write italic Pseudomonas aeruginosa in all manuscript text.

Reply: We have fixed Pseudomonas aeruginosa to italics throughout the manuscript text.

  1. Line 43 and resistance against antimicrobial agents correct to and resistance to antimicrobial agents

Reply: Changes made as suggested. Line 46.

  1. Line 93 Increasing the concentration of any DHP increased the inhibition of GFP production via the Las pathway. Correct to, Increasing the concentration of any DHP compounds increased the inhibition of GFP production via the Las pathway.

Reply: Changes made as suggested. Lines 101-102.

  1. Line 96 Figure 1. DHP compounds inhibit LasR-mediated production of GFP in aeruginosa MH602. Both halogenated and non-halogenated DHP compounds showed a concentration-dependent decrease in LasR receptor-mediated GFP production over 10 hours of growth. Peak 98 GFP production for all growth conditions occurred between 4-7 hours. At the 4-7 hours point even 99 in lowest concentration (12.5 μg/ml) of all DHPs produced a significant decrease (P < 0.05) in fluorescence compared to the control was observed. At 50 μg/ml, all compounds showed a complete 101 decrease in GFP fluorescence at 4-7 hours point. Data represent the average from three biological 102 replicates. Correct to  Figure 1. Showed effects of DHP compounds on GFP production in P. aeruginosa MH602, and write  P. aeruginosa with the same font size until matched with line. Add paragraph  Both halogenated and non-halogenated DHP compounds showed a concentration-dependent decrease in LasR receptor-mediated GFP production over 10 hours of growth. Peak 98 GFP production for all growth conditions occurred between 4-7 hours. At the 4-7 hours point even 99 in lowest concentration (12.5 μg/ml) of all DHPs produced a significant decrease (P < 0.05) in fluorescence compared to the control was observed. At 50 μg/ml, all compounds showed a complete 101 decrease in GFP fluorescence at 4-7 hours point. Data represent the average from three biological 102 replicates.  To results

Reply: Changes made as suggested. The results text is edited in order to avoid the repetition of sentences. Results have now been placed in Sub-section 2.2, and this Figure has been re-numbered as Figure 2 and Figure 2 legend.

  1. Line 118 Figure 2. DHP compounds inhibit GFP produced via the PqsR receptor protein in aeruginosa PAO1. Both halogenated and non-halogenated DHP compounds showed a concentration 119 dependent decrease in PqsR receptor mediated GFP over the 10 hours of growth. The GFP fluorescence peak for all growth conditions was between 4-7 hours point. At 4-7 hours point, even in the 121 lowest concentration of DHP (12.5 μg/ml), a decrease in fluorescence was recorded in comparison 122 to the control. Whereas, at 25 and 50 μg/ml all compounds showed a significant decrease (P < 0.05) 123 in GFP fluorescence at 4-7 hours point. Data represent the average from three biological replicates. . Correct to  Figure 2. Showed effects of DHP compounds on  GFP production in P. aeruginosa PAO1. Added paragraph Both halogenated and non-halogenated DHP compounds showed a concentration 119 dependent decrease in PqsR receptor mediated GFP over the 10 hours of growth. The GFP fluorescence peak for all growth conditions was between 4-7 hours point. At 4-7 hours point, even in the 121 lowest concentration of DHP (12.5 μg/ml), a decrease in fluorescence was recorded in comparison 122 to the control. Whereas, at 25 and 50 μg/ml all compounds showed a significant decrease (P < 0.05) 123 in GFP fluorescence at 4-7 hours point. Data represent the average from three biological replicates to results.

Reply: Changes made as suggested. The results text is edited in order to avoid the repetition of sentences. Results have now been placed in Sub-section 2.3, and this Figure has been re-numbered as Figure 3 and Figure 3 legend.

  1. The authors used Green Fluorescent Protein (GFP) to determine the inhibition level of LasR and PqsR with the same methodology. How do authors discriminate between LasR and PqsR?

Reply: To identify which QS receptor protein (LasR or PqsR) is inhibited, we used two separate isolates that were tagged with GFP protein. MH602 is tagged only for LasR and PAO1-PQS is tagged for PqsR only.  This information is already mentioned in Materials and Methods, Sub-section 4.2 Lines: 302-303.

  1. In figure3A growth of effect of W1- aeruginosa growth reference strain growth correct to  Figure3A effects of DHP on P. aeruginosa growth (reference strain)

Reply: Changes made as suggested. Refer to the re-numbered Figure (4A).

  1. In figure 3B growth of effect of W1- aeruginosa growth reference strain growth correct to  Figure3B effects of DHP on P. aeruginosa growth (clinical strain)

Reply: Changes made as suggested. Refer to the re-numbered (Figure 4B).

  1. Line 125 DHP compounds have minimal impact on P. aeruginosa growth correct to Assess effects DHP compounds on aeruginosa.

Reply: Changes made as suggested. Line. 126

  1. Title figure 3 correct to Figure 3 Effects of DHP compounds on aeruginosa growth and biofilm biomass.  Paragraph  DHP compounds at any concentration 140 did not significantly affect the growth of any strain (A and B). In presence of DHP compounds, biofilm formation was 141 significantly reduced for all isolates except for DFU-53 which was only significantly affected by compound 4a after 48 142 hours of biofilm formation (C). Growth data is representative of the average from three biological replicates. Biofilm bio-143 mass data is representative of the average from four biological replicates * indicates differences are statistically significant 144 P < 0.05. added to results  

Reply: Changes made as suggested. The results text is edited in order to avoid the repetition of sentences. Results – currently Sub-section 2.4 and 2.5. and re-numbered, Figure 4 legend.

  1. Line 146 Biofilm formation is hindered in the presence of DHP correct to Effects of DHP on biofilm formation by aeruginosa

Reply: Changes made as suggested. re-numbered Line 154.

  1. Line 154 DHP impedes pyocyanin production of P. aeruginosa isolates correctly to DHP impedes pyocyanin production by P. aeruginosa strains

Reply: Changes made as suggested. re-numbered Line 175.

  1. Line 286 clinical strains correct to clinical isolates

Reply: Changes made as suggested. re-numbered Line 303.

  1. Title figure 5 Figure 4. DHP compounds inhibit pyocyanin production by aeruginosa isolates. Both 173 halogenated and non-halogenated DHP compounds showed concentration-dependent decreases 174 in pyocyanin production after 48 hours of growth. At 50 μg/ml all compounds showed a signifi-175 can’t decrease. Data represent the average from n=3 biological replicates. * Indicates differences are 176 statistically significant P < 0.05. correct to Figure 4. Showed effects of DHP on pyocyanin production by P. aeruginosa strains and write with one font size. Paragraph  Both halogenated and non-halogenated DHP compounds showed concentration-dependent decreases in pyocyanin production after 48 hours of growth. At 50 μg/ml all compounds showed a significant decrease. Data represent the average from n=3 biological replicates. * Indicates differences are 176 statistically significant P < 0.05 added to results.

Reply: Changes made as suggested. The results text is edited in order to avoid the repetition of sentences. Results – currently Sub-section 2.6. and re-numbered, Figure 5 legend.

  1. Line 178 Bromophenyl-DHP analogue (4a) shows concentration-dependent cytotoxicity to HFF-1 cell correct to Cytotoxicity Study of -DHP analogue (4a) on human foreskin fibroblast (HFF-1) cell.

Reply: Changes made as suggested. Line 196.

  1. Paragraph Microscopy images showing HFF-1 cell 190 morphology and confluence at 24 hours (A). At the low and moderate concentrations (12. 5 and 191 μg/ml), 4a was not cytotoxic but at 50 μg/ml the viability of HFF-1 reduced to approximately 67% 192 (B). Cell viability is representative of the average from three biological replicates. * Indicates differences were statistically significant P < 0.05. deleted from the title of figure 5 and added to results.

Reply: Thanks for the suggestion, we have now removed the text from the title and added it to the results. Sub-section 2.7.

  1. Line 289 All clinical isolates were de-identified before being gifted to us from the hospitals. None of the authors were involved in the isolation or characterization of clinical isolates from 290 patients. The must be mentioned who is responsible for the identification of bacterial isolates and the method used.

Reply: Clinical bacteria from patients were previously isolated and characterized by qualified pathologists from the respective hospitals based on procedures detailed in “ASM clinical Microbiology procedures handbook-4th Edition, ASM Press (edited by Amy L. Leber)”. We have now added this information in the methods sub-section 4.2. Lines 306-308.

To Note: The hospital pathologists are not part of this study nor included in the author list.

  1. Line 291 authors mention, All isolates were stored at -80ºC in 25% DMSO and used as necessary by plating, How bacteria are stored in 25% DMSO? Mention source this is information?

Reply: Bacterial isolates once received in our lab were immediately grown in a full-strength (100%) Tryptone Soy Broth (Oxoid, ThermoFisher, NSW, Australia) in a test tube at 37ºC, 150 rpm for 24 hours. Following 24 hours of growth, the bacterial stock was prepared by mixing 1.2 ml of bacterial culture with 400 µL of DMSO (100%), and then the bacterial stock was stored in 2 ml of the cryogenic vials (Corning, Sigma-Aldrich, NSW, Australia) at -80ºC. To note: the final concentration of DMSO in the bacterial stock is 25%. P. aeruginosa isolates from the frozen stock were plated as required onto Tryptone Soy agar (Oxoid, ThermoFisher, NSW, Australia) and grown for 24 hours by incubating at 37ºC. Further growth was initiated by striking a colony from the Tryptone Soy agar plate into the 5 ml of Muller–Hinton Broth (MHB, Oxoid, ThermoFisher, NSW, Australia) for all experiments. This information has been added in the methods subsection 4.2.

  1. In the statical analysis part, the authors said, For all statistical analysis, Graphpad Prism 384 was used for two-tailed unpaired T-test to determine the P-values. Why did not the authors perform an ANOVA analysis to compare the various compound at the level of each effect (growth quantification assay, biofilm biomass quantification, pyocyanin production, etc.) and therefore choose the appropriate molecule to employ the cytotoxicity experiment?

Reply: Two-tailed unpaired T-tests to determine the P-values is a very common and highly acceptable statistical method. From our results, it is very clear that all compounds showed a statistically significant difference in comparison to the untreated control. However, compounds 4a (Br) and 4c (F) have shown the best efficacy specifically with inhibition of pyocyanin production (Figure 5) and inhibition of PqsR receptor responsible for pyocyanin biosynthesis (Figure 3). For the cytotoxicity study, in addition to compound 4a (Br), we have now also included compound 4c (F) at 25 and 50 µg/ml. Refer to Figures 6A and B. Both compounds 4a and 4c have similarly low cytotoxic effects on HFF-1 cells. Addition data have been included in the Results section (2.7), Discussion, and Materials and Methods section (4.7).

Round 2

Reviewer 1 Report

Acceptable in its current form

Author Response

Thank You for reviewing our manuscript.

Reviewer 2 Report

  1. Figure (1) is not correct and need to modification.
  2. aeruginosa must be written italic on all manuscript text line 306, 327, 358.

Author Response

We thank the reviewer for their comments and suggestion. We have considered the reviewer’s suggestions and made all changes. All changes in the manuscript are highlighted in yellow

  1. Figure (1) is not correct and need to modification.

Reply: We apologize, for the mistake. We noticed Y-axis and X-axis title is missing in Figure -1. We have now added it to the figure and changed bar colours for clear differentiations. Also, Figure 1 legend changed to “The susceptibility of P. aeruginosa isolates to ciprofloxacin.”

Figure 1. The susceptibility of P. aeruginosa isolates to ciprofloxacin.

  1. P. aeruginosa must be written italic on all manuscript text line 306, 327, 358.

Reply: We checked the whole text in the manuscript including supplementary and changed P. aeruginosa to italics.
